# The Incorporation of Carvacrol into Poly (vinyl alcohol) Films Encapsulated in Lecithin Liposomes

**DOI:** 10.3390/polym12020497

**Published:** 2020-02-24

**Authors:** Johana Andrade, Chelo González-Martínez, Amparo Chiralt

**Affiliations:** Instituto Universitario de Ingeniería de Alimentos para el Desarrollo, Universitat Politècnica de València, Camino de Vera s/n, 46022 Valencia, Spain

**Keywords:** food packaging, encapsulation, PVA, degree of hydrolysis

## Abstract

Lecithin-encapsulated carvacrol has been incorporated into poly (vinyl alcohol) (PVA) for the purpose of obtaining active films for food packaging application. The influence of molecular weight (Mw) and degree of hydrolysis (DH) of the polymer on its ability to retain carvacrol has been analysed, as well as the changes in the film microstructure, thermal behaviour, and functional properties as packaging material provoked by liposome incorporation into PVA matrices. The films were obtained by casting the PVA aqueous solutions where liposomes were incorporated until reaching 0 (non-loaded liposomes), 5 or 10 g carvacrol per 100 g polymer. The non-acetylated, high Mw polymer provided films with a better mechanical performance, but less CA retention and a more heterogeneous structure. In contrast, partially acetylated, low Mw PVA gave rise to more homogenous films with a higher carvacrol content. Lecithin enhanced the thermal stability of both kinds of PVA, but reduced the crystallinity degree of non-acetylated PVA films, although it did not affect this parameter in acetylated PVA when liposomes contained carvacrol. The mechanical and barrier properties of the films were modified by liposome incorporation in line with the induced changes in crystallinity and microstructure of the films.

## 1. Introduction

Packaging is necessary for preserving food quality and extending its shelf life [1], but the accelerated generation of non-biodegradable plastic wastes, many of them from the food packaging sector, has generated a serious environmental problem that impacts consumer awareness. Many efforts have been directed towards the search for materials based on biodegradable polymers and the development of new packaging models to address this issue. Of these, the use of biodegradable materials with antimicrobial and antioxidant properties represents innovation in the concept of food packaging [2]. These materials seek to reduce the environmental impact, while also ensuring the food quality and safety.

Poly (vinyl alcohol) PVA is a biodegradable synthetic polymer that was obtained from the controlled hydrolysis of polyvinyl acetate (PVAc), with high (≈99%), medium (≈88%), or low (≈78%) hydrolysis degree [3]. The use of this polymer for food packaging purposes could be of great interest, as it is transparent, non-toxic, odourless and water-soluble, while exhibiting good mechanical properties [4]. The degree of hydrolysis and the molecular weight are relevant molecular parameters in the polymer functionality, since these affect the physical properties, such as viscosity and film-forming capacity, as well as the elasticity and tensile strength of their films [5]. These molecular characteristics also affect the polymer affinity and compatibility with other compounds that can be added as active components [6], which makes the polymer more or less appropriate for the development of active films.

Several compounds of natural origin, such as essential oils and plant extracts, which exhibit antimicrobial and antioxidant properties, can be used in the development of active materials [7]. Carvacrol is a monoterpenoid phenol that is found in the essential oil of oregano (Origanum vulgare), thyme (*Thymus vulgaris* L.), marjoram (Origanum majorana), and similar aromatic plants [8], which is widely studied for its outstanding antimicrobial [9] and antioxidant action [10].

Essential oils or their components have been emulsified in the polymer aqueous solutions to obtain active films from water soluble polymers by casting [11]. However, this process usually involved significant losses of the volatile compounds, mainly during the film drying step. In this step, emulsified essential oils are susceptible to destabilization processes, such as droplet flocculation, coalescence, and creaming, which lead the drops to the surface of the film where the lipid compounds are lost by steam drag effect along with evaporated water [12,13]. Therefore, the active capacity of the films is significantly reduced in line with the decrease in the retention capacity of the active components [14]. The use of nano/micro-encapsulation techniques to entrap these volatile compounds could mitigate this problem [15,16].

Liposomes are phospholipid vesicles that are organized in one or several concentric bilayers, with an aqueous inner core, which is able to self-assemble in aqueous solutions by hydrophobic effects [17]. They represent an efficient approach for encapsulating essential oils, thus improving their solubility and chemical stability in water systems [18,19,20], also allowing their more controlled release [21,22]. Liposomal encapsulation has been proved to enhance the retention of different essential oils during film formation by casting [15,23]. However, the inclusion of liposomes in the different polymeric matrices also generated changes in the microstructure of the films. The changes in the solvent properties during the film drying step alter the vesicular structure due to the liotrophic mesomorphism of polar lipids. Previous studies reported the formation of lamellar lipid associations in starch-gellan [15] and chitosan films [23] with incorporated lecithin liposomes that were loaded with essential oil compounds. These changes at the microstructural level promoted modifications in the film functional properties. Nevertheless, liposomes notably improved the retention capacity of the active compounds in the film matrix. 

In this study, carvacrol encapsulated in lecithin liposomes was incorporated into two types of poly (vinyl) alcohol (PVA), in order to analyse the influence of the molecular weight (Mw) and the degree of hydrolysis of PVA on the ability to form active films. To this end, the microstructural, thermal, mechanical, and barrier properties of the films with encapsulated carvacrol were evaluated.

## 2. Materials and Methods

### 2.1. Materials 

Two kinds of poly (vinyl) alcohol (PVA) with different molecular weight and degree of hydrolysis (A: Mw 89,000–98,000; 99-99.8% hydrolysed and B: Mw 13,000–23,000; 87–89% hydrolysed), carvacrol (CA) (Sigma-Aldrich, Steinheim, Germany), and soy lecithin (L) (Lipoid S75, Lipoid GmbH, Ludwigshafen, Germany), containing 72% phosphatidylcholine, 10% phosphatidulethanolamine and 2% lysophosphatidylcholine, were used in the preparation of the films. Magnesium nitrate (Mg(NO_3_)_2_), phosphorus pentoxide (P_2_O_5_) salts, and UV-grade methanol were supplied by Panreac Química S.A. (Barcelona, Spain).

### 2.2. Preparation of Liposome Dispersions

Lecithin was initially dispersed in distilled water (5% w/w) while using magnetic stirring for 15 min at 800 rpm. Afterwards, the dispersion was subjected to sonication (35 kHz) for 10 min with pulses of 1 s, by using an ultrasonic device (Vibra Cell, Sonics & Material, Inc., Newtown, CT, USA). In this step, carvacrol (2.5 or 5% w/w) was incorporated into the lecithin dispersions. Thus, three liposome dispersions were obtained: one containing non-loaded liposomes (L) and two carvacrol loaded (L-CA5, L-CA10). The dispersions were placed in an ice bath during the ultrasound application to prevent sample heating.

### 2.3. Preparation of Films 

The PVA films were obtained by casting the PVA aqueous solutions. Polymer solutions (A 5% wt. and B 10% wt.) were prepared in distilled water while using magnetic stirring (1,200 rpm) at 100 °C for 3 h. Liposome dispersions (L, L-CA5, L-CA10), were added to the polymer solutions to reach a lecithin:polymer ratio of 10% with variable ratios of CA (0%, 5% or 10% wt. with respect to the polymer). The control films were obtained with the pure polymer solutions. All of the formulations were degassed by using a vacuum pump and spread evenly onto Teflon plates of 150 mm in diameter, while using a constant equivalent mass of polymer per plate of 2 g. The films were dried under controlled temperature (25 ± 2 °C) and relative humidity (54 ± 2%) for 48 h. Table 1 shows the different film formulations with the respective mass fractions of the components.

The films were conditioned for one week at 53% relative humidity (RH), while using Mg(NO_3_)_2_ over-saturated solution, before their characterisation. Meanwhile, the final CA content in the film (CA retention), microstructure, and thermal analyses were carried out with films conditioned at 0% RH using P_2_O_5_.

### 2.4. Characterization of the Active Poly (vinyl alcohol) Films

#### 2.4.1. CA Retention and Structural Arrangement

The determination of CA that was retained in the different formulations after film drying was carried out by the total extraction of CA from the films and its spectrophotometric quantification. The film samples (4 cm^2^) were immersed in 50 mL of a 50% aqueous solution of UV-grade methanol, and then kept under stirring (300 rpm) for 48 h at 25 °C. Aliquots of the samples were extracted and the absorbance (A) was measured at 274 nm, by using a spectrophotometer (Evolution 201 UV-Vis, Thermo Fisher Scientific Inc., Waltham, MA, USA). The CA concentration (C) in the films was determined by means of a standard curve, which was obtained while using solutions containing between 10 and 50 µg/mL of carvacrol in the same solvent (C = 63.61A, R^2^ = 0.998). The backgrounds used for the measurements were the corresponding extracts that were obtained under the same conditions from the CA-free films. CA retention (%) in the films was calculated as the ratio between the mass of CA extracted from the film with respect to the corresponding mass of CA initially incorporated.

The structural arrangement was evaluated in the cross-sections of the films, by using a Field Emission Scanning Electron Microscope (FESEM) (ZEISS®, model ULTRA 55, Oberkochen, Germany), at an acceleration voltage of 2 kV. The film samples were cryofractured by immersion in liquid nitrogen and then coated with platinum before obtaining the images. 

A Nicolet 5700 spectrometer (Thermo Fisher Scientific Inc., Waltham, MA, USA) was used to obtain the FTIR spectra of the film samples. The average spectra were collected from 64 scans with a resolution of 4 cm^−1^ in the 4000–400 cm^−1^ range. These were performed in triplicate and at three different locations in each sample. 

The X-ray diffraction spectra of the films were recorded with a D8 Advance X-ray diffractometer (Bruker AXS, Karlsruhe, Germany) that was in the range 2θ: 10° and 50°, with a step size of 0.05, while using Kα Cu radiation (λ: 1.542 Å), 40 kV and 40.mA. The diffraction curves were deconvoluted by using the Lorentz model and the OriginPro 8.5 software to obtain crystalline and amorphous regions. The degree of crystallinity of the samples (Xc, expressed as a percentage) was estimated from the ratio of crystalline peak areas and the total area of the original diffractograms.

#### 2.4.2. Thermal Behaviour

The thermal behaviour of the films was assessed by thermogravimetric analysis (TGA) and differential scanning calorimetry (DSC). TGA analysis was performed by heating the samples in alumina crucibles from 25 °C to 700 °C at 10 °C/min. under a 10 mL/min. nitrogen stream, by using (TGA/SDTA 851e, Mettler Toledo, Schwarzenbach, Switzerland). The DSC measurements of samples in aluminium crucibles were carried out under nitrogen flow (10 mL/min.) while applying heating (first and second) and cooling scans in a differential scanning calorimeter (DSC 1 stareSystem, Mettler Toledo, Schwarzenbach, Switzerland). The temperature scanning profile was: a first heating from −25 °C to 250 °C at 10 °C/min., holding for 2 min. at 250 °C, sample cooling to −25 °C, holding for 2 min. at −25 °C; and, a second heating from −25 to 250 °C at 10 °C/min. All of the measurements were taken in triplicate.

#### 2.4.3. Functional and Optical Properties

The tensile properties of the films (elastic modulus (EM), tensile strength (TS), and elongation at break point (E)) were determined from the tensile stress (σ) vs. Hencky strain (εH) curves, following the standard method ASTM D882-02 [24]. The test specimens (25 mm x 100 mm) were mounted in the film extension grips of the universal testing machine (Stable Micro Systems, TA.XT plus, Haslemere, England); the initial separation of the clamps was 50 mm and stretched at 50 mm.min.^−1^ until breaking. The measurements were carried out in eight samples for each treatment.

Water vapour permeability (WVP) was analysed following a modification of the E96/E95M-05 gravimetric method [25]. The film samples of each formulation were placed on Payne permeability cups 3.5 cm in diameter (Elcometer SPRL, Hermelle/s Argentau, Belgium) at 25 °C and 53–100% RH gradient, which was created with an oversaturated Mg(NO_3_)_2_ solution (inside the desiccator where the cups were placed) and distilled water (5 mL, inside the cup). A fan was positioned above each cup in order to reduce the resistance to transport of water vapour. The cups were weighed periodically every 1.5 h for 24 h while using an analytical balance (±0.00001 g). The slopes of the weight loss vs. time at the steady state period were determined by linear regression to calculate the water vapour transmission rate (WVTR). WVP was calculated as described by Cano et al. [26]. For each type of film, WVP measurements were carried out in triplicate.

The oxygen permeability (OP) was analysed in film samples (50 cm^2^) by using an Ox-Tran system (Mocon, Minneapolis, US) at 23 °C and 53% RH, following a standard method F1927-07 [27]. The films were exposed to pure nitrogen flow on one side and pure oxygen flow on the other side. The oxygen transmission rate was multiplied by the average film thickness and divided by the partial pressure of oxygen to calculate the OP. Each film formulation was analysed in triplicate. The film thickness was measured with a digital electronic micrometer (Palmer, COMECTA, Barcelona, Spain) to the nearest 0.001 mm at six random positions. Ten measurements were taken for each sample to determine the average.

The optical properties (transparency and colour coordinates) were determined by using a spectrocolorimeter (CM-3600d Minolta CO., Tokyo, Japan) to obtain the reflectance spectra of the samples from 400 to 700 nm, on white (R) and black ($R_{0}$) backgrounds, as well as the reflectance of the white backing ($R_{g}$). The transparency was measured through the internal transmittance (Ti), while applying the Kubelka-Munk theory for multiple scattering (Equation (1)). The CIE L*a*b* colour coordinates were obtained from the reflectance of an infinitely thick film (R_∞_) (Equation (2)) spectra, using the 10° observer and the D65 illuminant as reference, according to Hutching [28]. Psychometric coordinates, Chroma (C_ab_*) and hue (h_ab_*) were also determined while using Equations (5) and (6). Three measurements were taken from each film and three films were considered per formulation.
(1)$$T_{i}=\sqrt{{\left(a\;+R_{0}\right)}^{2}-b^{2}}$$
(2)$$R_{\infty }=a-b$$
(3)$$a=\frac{1}{2}\;\left[R+\;\frac{R_{0}-R+R_{g}}{R_{0}\times R_{g}}\right]$$
(4)b= a2−112
(5)$$C_{ab}^{*}=\sqrt{\left(a^{*2}+\;b^{*2}\right)}$$
(6)$$h_{ab}^{*}=arctg\;\frac{b^{*}}{a^{*}}$$

#### 2.4.4. Statistical Analysis

The statistical analysis of the data was carried out while using Statgraphics Centurion XVI.II. The results were submitted to an analysis of variance (ANOVA). Fisher’s least significant difference (LSD) was used at the 95% confidence level. 

## 3. Results and Discussion

### 3.1. Microstructure 

Figure 1 shows the FESEM micrographs of the cross-section of the different PVA films (A and B). Homogeneous PVA films were obtained with both polymers, A and B, but a dispersed phase in the polymer matrix can be observed for formulations containing lecithin liposomes loaded, or not, with carvacrol. This dispersed phase corresponds to the different incorporated lipid materials (lecithin liposomes with and without carvacrol), which are not miscible with the polymer. The lipid particles are larger than the liposomes initially incorporated (180 nm) into the aqueous phase, which indicates that liposomes, either loaded or not with carvacrol, changed during the film formation. Throughout the film drying step, the evaporation of the solvent, where the initial liposomes were dispersed, provoked changes in the liposomal structure due to the modification of the lipid-solvent interactions, thus giving rise to different mesomorphic species (lyotropic mesomorphism) containing lecithin components and carvacrol that remain dispersed in the polymer matrix, as previously observed by other authors [15,29]. The carvacrol that was released from liposomes could partially evaporate during the film drying step in line with the coalescence and creaming of the formed droplets and steam drag effect on the film surface [30]. 

The volume fraction of the particles and the particle size distribution were different for a determined lipid composition in polymers A and B, as shown in Figure 1. A higher proportion of dispersed lipid particles was observed in the polymer A matrix, where the dispersed particles were smaller for the highest ratio of carvacrol. In contrast, matrix B exhibited a lower ratio of dispersed phase, with particle size also being affected by the carvacrol ratio (0, 5 or 10 g/100 g polymer) with a constant amount of lecithin. For non-loaded liposomes, the particle size was quite homogenous and bigger than that of carvacrol-loaded liposomes, for which a higher proportion of smaller particles could be observed. These microstructural differences between films A and B could be attributed to the molecular differences in the polymer chains. While polymer A is highly hydrophilic with a high number of hydroxyl groups, the presence of residual acetyl groups in the polymer B chains provides them with an amphipathic nature [31] that promotes the chemical affinity between the polymer and the lipid compounds. This greater affinity enhances the molecular integration of the lipids in the matrix, thus giving rise to a smoother and more homogeneous microstructure than that obtained with A-PVA. 

In both polymers A and B, the highly-loaded carvacrol liposomes seem to have greater stability, remaining smaller in the dry film, which was probably due to the better structural cohesion of the liposomal membrane promoted by carvacrol. Other authors [32,33] also observed greater stability of the lipid associations with carvacrol by analysing liposomal systems by means of epifluorescence imaging and H-NMR spectroscopy techniques. These authors reported that the spatial rearrangement of carvacrol in the liposomes resulted in the reduction of repulsive forces among the head groups of phospholipids, which decreased the mobility degree of hydrocarbon chains, thus allowing for closer molecular packing. However, liposomes, carvacrol-loaded or not, generated subtle structural changes in the PVA B films, regardless of the carvacrol ratio. This could be explained by the greater chemical affinity of carvacrol with the polymer that can enhance its bonding to the polymer chains, limiting its interactions with the lecithin structures, thus giving rise to a partition of the compound between the polymer chains (bonded fraction) and lecithin liposomes (dispersed fraction). In fact, matrices of PVA B containing different ratios of free carvacrol did not exhibit a dispersed phase, showing a homogeneous structure [34]. Subsequently, the lipid particles dispersed in PVA B matrices must be mainly attributed to the lecithin lipid associations coming from the destructuring of incorporated liposomes during the film drying step, which can also entrap a part of the carvacrol. Afterwards, a partition of carvacrol between the polymer chains and lecithin lipid associations could be assumed in the PVA B matrices, which would explain the microstructural differences between samples A and B. 

Table 1 shows the mass fraction of the different components of the films and the carvacrol concentration in the matrix as well as the percentage of CA retention (with respect to what was initially incorporated) in the final film. The specific molecular characteristics of each type of PVA generated different CA-retention levels. Matrix B retained a significantly (*p* < 0.05) higher amount of CA (67% and 74%) than matrix A (55% and 57%). This has also been observed for emulsified carvacrol in PVA A and B films and it was attributed to the greater chemical affinity of CA with PVA B [34]. The formation of Lewis adducts between CA and the residual acetate groups of PVA B chains has been reported to describe the specific molecular interactions. When comparing the amount of carvacrol retained in the films by liposome encapsulation with that obtained by direct emulsification [34], an increase of about 15% in carvacrol retention was observed when using liposomes as carvacrol carriers. This fact indicates that liposomal encapsulation enhances the retention of non-polar active compounds, as carvacrol, in hydrophilic polymer matrices that were obtained by casting the aqueous polymer dispersions, as has also been observed by other authors [15,23,35]. Subsequently, although the liposomes lost the initial structure, the modified structures also contributed to stabilizing the non-polar compounds (such as carvacrol) against flocculation, coalescence, and creaming, which are mainly responsible for their losses at the film surface in line with water evaporation.

The crystallization pattern and degree of crystallinity of the films were analysed through the DRX spectra (Figure 2), all of which showed the typical PVA crystalline peaks at around 2Ө of 20° and 41°. The crystallinity of films A (51%) was higher than that of films B (45%), as previously described by [34], which was attributed to the presence of acetyl groups in the B chains that provoke a steric hindrance for the crystalline molecular arrangement. The incorporation of liposomes (loaded or not with CA) led to a significant decrease in the crystalline fraction of matrix A. This could be due to the presence of the dispersed lipid phase that might disrupt the normal organization of polymeric chains. The AL-CA5 samples had the lowest crystallinity value (25%), in line with the more heterogeneous distribution of lipid fraction in the matrix (Figure 1), as previously commented on. On the other hand, even though the non-loaded liposomes promoted a drop in crystallinity of films B (35%), the liposomes containing CA gave rise to films with crystallinity that was similar to that of pure polymer films. Andrade et al. [34] observed an increase in the crystallinity of PVA B films when these contained carvacrol, which was attributed to the specific interactions of the carvacrol molecule and PVA chains that could favour the molecular arrangement in the crystalline domains. Subsequently, the decrease in crystallinity provoked by the lecithin incorporation was less significant when liposomes carried carvacrol, whose delivery and interaction with the polymer chains enhance crystallinity. The different crystallinity degree of the films affected their functional properties, as described below.

The molecular interactions between loaded compounds and polymer chains could lead to molecular vibration changes that are reflected in the FTIR spectra. The FTIR spectra of all PVA films (Figure 3) showed the characteristic bands of PVA; the strong broadband between 3600 cm^−1^ and 3000 cm^−1^ corresponding to the stretching vibration of hydroxyl (O-H) and the intermolecular hydrogen bond. Two peaks that appear between 3000 cm^−1^ and 2800 cm^−1^ are related to the -CH_2_ asymmetrical and symmetrical stretching vibration of the alkyl groups. Other peaks that were related to the -CH_2_ bending (1416 cm^−1^), the deformation of C-H vibrations (1326 cm^−1^), C-O stretching (1085 cm^−1^), and C-O-C ring vibration (832 cm^−1^) were also observed due to the motion of the carbon skeleton of PVA [6,36]. In matrix B, two additional peaks that were associated with stretching vibrations of the ester group and the carbonyl (C=O) group stretching (corresponding to the residual acetyl groups) were observed at 1240 cm^−1^ and 1732cm^−1^ [37,38,39,40]. 

The incorporation of liposomes in films A led to the appearance of characteristic peaks of phosphatidylcholine corresponding to the asymmetrical and symmetrical CH stretching at 2920 cm^−1^ and 2850 cm^−1^, C=O stretching at 1730 cm^−1^, and C=C stretching vibration at around 1240 cm^−1^ [41,42]. In films of PVA B, the characteristic peaks of phosphatidylcholine are overlapped with those of the polymer. The typical FTIR peaks of carvacrol have been observed by other authors [37,38,39,40] at 3500–3300 cm^−1^ (-OH stretch), 2868–2958 cm^−1^ (C-H stretch), 1620^–^1485 cm^−1^ (C-C stretch), 1240 cm^−1^ (C-O stretching vibration in aromatic ring), and 800 cm^−1^ (aromatic C-H bending). In no case did the presence of CA cause changes in the vibration band of PVA. The described interaction between CA and acetyl groups in PVA B chains did not cause changes in the vibration mode of carbonyls. This can be attributed to both the very low molar ratio of CA in the films, which prevents the quantitative observation of its characteristic vibration bands, and the lack of covalent bonds between CA and PVA groups.

### 3.2. Thermal Behaviour

The alterations that the heating can cause in the materials, such as dehydration, oxidation, combustion, and decomposition, were studied through thermogravimetric analysis (TGA). Figure 4 shows the TGA and DTGA (first derivate) curves, which showed the different weight loss steps of material as a function of temperature. The first step, between 50 °C and 120 °C, corresponds to the sample water loss due to the vaporization of the bonded water of the polymer matrix. Every sample of polymer A, including those with liposomes, had 3% of bonded water. In contrast, only 1.6% bonded water was determined in PVA B films, which was in agreement with its more hydrophobic nature, associated with the acetylated hydroxyls. The incorporation of carvacrol into films B, and the developed interactions with the polymer, also reduced the polymer bonded water to 0.6%. A second weight loss step that only appeared in samples containing carvacrol is attributable to this compound thermo-release. In this step, the peak temperatures were roughly at 196 °C and 150 °C for A and B films, respectively. The different temperature in each polymer could be attributed to their different melting behaviour, as described below, since carvacrol will be effectively delivered from the polymer melt. The integration of the DTGA peak of carvacrol evaporation permits the estimation of the amount of carvacrol that was thermo-released from the A and B matrices. The estimated amounts were 60–70% from the compound that was retained in the case of matrix A (poorer in carvacrol) and 33–46% in matrix B (richer in carvacrol). Therefore, matrix B, with a greater carvacrol affinity, allowed for a better carvacrol retention in the films with more thermal stability. 

At higher temperatures, the polymer degradation occurred in two different steps; the first was associated with the detachment of side groups from the chains, forming water in the case of polymer A and acetaldehyde and acetic acid, as additional by-products, in polymer B with acetylated hydroxyls. The second stage of polymer degradation has been related to the degradation of low molecular weight products from the decomposition of the main chain, or of heavier structures that formed in the previous degradation steps [43]. In polymer A, the first degradation step occurred between 219 °C and 333 °C, exhibiting two peaks that indicates the overlapping of different weight loss events with the respective degradation mechanisms. This could be related with the fact that the melting of the crystalline fraction (Tm: 225 °C) and degradation both occurred in the same temperature range, generating fractions in different physical states that should degrade differently [34]. The incorporation of liposomes into the PVA matrix provoked a shift in the degradation temperature to higher values (onset at 255 °C), thus increasing the thermal stability of the polymer matrix, while only one degradation peak temperature was observed between 297 °C and 314 °C. The thermo-protection effect of lecithin in PVA A films could be due to the presence of carbonyl from lecithin. Some authors [43,44] reported the thermo-protective effect of carbonyls in PVA degradation, due to the increase in the value of the activation energy that was associated with the material degradation. In fact, for the partially acetylated PVA B, the polymer degradation occurred at higher temperatures in a single degradation peak at 281 °C. Moreover, degradation occurred after the polymer melting at 185 °C. 

The incorporation of liposomes into PVA B matrices also implied an increase in the degradation peak temperature of the polymer (302 °C), although the mass loss step started at 210 °C (as compared to 230 °C in lecithin-free matrices). Figure 4 also shows the wide thermo-degradation curve of lecithin with overlapped multiple peaks that were associated with the degradation of its different components (phosphatidylcholine (75%), phosphatidylethanolamine (10%), lysophosphatidylcholine (2%), and free fatty acids, triglycerides). At 210 °C, the quantitative degradation of the lecithin present in the films, which overlapped with the polymer degradation, can cause the apparent onset at 210 °C for polymer B degradation. Accordingly, lecithin provides thermo-protection to both A and B polymers in the obtained films probably due to the presence of carbonyls in the blend. The temperature peak for the second stage of polymer degradation was above 400 °C for both A and B matrices and also slightly shifted towards higher temperatures when liposomes were incorporated. Moreover, there was a significant change in the percentage of mass loss that was associated with the two different degradation stages of polymer A. The presence of lecithin provoked an increase in the mass loss of the second step with respect to the first, which could be due to the interactions that developed between lipids and polymer during heating that alter the degradation pattern of the PVA. Accordingly, lecithin provided greater thermal stability to PVA, which is particularly relevant for non-acetylated PVA with overlapped melting and degradation events.

The PVA is a semi-crystalline polymer with amorphous and crystalline fractions, which is reflected in first and second order transitions in the DSC thermograms (Figure 5). Table 2 shows glass transition and melting temperatures determined from the thermograms corresponding to the first and second heating scans. The former reflects the thermal behaviour of the cast films, while, in the second heating scan, the polymer melting in the previous heating step has erased the thermal history. In the first heating, a relaxation enthalpy that is associated with the glass transition can be observed as a result of molecular relaxation after aging [45]. The glass transition temperature (Tg) in polymer A (46 ℃) was lower than in polymer B (54 ℃). This could be due to different factors, including the degree of hydrolysis; matrix A, with a high degree of hydrolysis (99%), has a greater number of hydroxyl groups available to bind water, whose plasticizing effect decreases the Tg. In the same sense, the presence of residual acetyl groups in partially hydrolysed chains of B, allows less water binding, as commented on above. Other factors that are associated with the film preparation process could also affect the Tg value. The distribution of polymer chains of different lengths in the amorphous or crystalline phases due to the crystallization restrictions that are associated with the steric hindrance could occur. Shorter molecular chains could be located in the amorphous phase of matrix A. 

The incorporation of liposomes with and without carvacrol slightly increased the Tg of polymer A, whereas the opposite effect was observed for carvacrol-loaded liposomes in polymer B. This suggests that the lecithin components had an anti-plasticizing effect in fully hydrolysed PVA, whereas the carvacrol, as released from liposomes, plasticized PVA B, as previously observed by Andrade et al. [34]. In the second heating scan, the loss of bonded water in the matrix and the different chain rearrangement after melting increased the Tg values in polymer A, especially in the samples with lecithin, which confirm its anti-plasticizing effect in PVA A. The Tg values of sample B in the second heating were only slightly higher than in the first, which could be mainly attributed to the loss of bonded water during the first heating, with the formation of an amorphous phase with a similar composition to that present in the cast films. The incorporation of lecithin into PVA B also had an anti-plasticizing effect after the first heating, but this effect seemed to be mitigated by the carvacrol plasticization when carvacrol was present. The apparent presence of carvacrol in the samples that were previously heated up to 250 °C suggests the thermo-protection of CA by liposomal encapsulation. A previous study [34] showed no evidence of CA after the sample heating up to 250 °C, when CA was incorporated in matrices A and B by direct emulsification.

DSC thermograms of the first heating scan (Figure 5 show several endothermic events between 140 °C and 250 °C for both A and B polymers. Polymer A presented peaks at 215 °C and 225 °C that suggest two successive melting events, which is in line with the existence of crystalline forms of different sizes leading to different melting temperatures. When liposomes were incorporated, the melting enthalpy increased, which suggests that lipid compounds could enhance the PVA crystallization in the smallest crystals, since it was the peak at the lower temperature that was promoted. The partial vaporization of CA thermo-released in this temperature range could also contribute to the enthalpy increase. Matrix B exhibited a melting endotherm at 183 ℃ with enthalpy values that were lower than matrix A. In this case, the incorporation of lipid compounds (lecithin and CA) did not modify the Tm value or the enthalpy. Nevertheless, a second endothermic peak was present in all of samples B above 230 °C, which must be attributed to the early endothermic degradation reactions (deacetylation) of the polymer that start at this temperature [43,46].

In the second heating scan, polymer A only presented one endothermic peak, at 225 °C, as previously reported by Cano et al. [6] and David Salazar [47], whereas polymer B exhibited the melting peak at 168 °C. The Tm value was significantly higher in matrix A, coherently with the differences between the polymers’ molecular weights and the steric effect of the side acetyl groups of B on the crystalline arrangement of the chains. Polymer A did not modify the Tm with respect to the previous step, which indicated a similar composition in its crystalline fraction even after heating the sample up to 250 °C. However, the presence of lipid compounds slightly reduced the peak temperature and enthalpy, which indicates that the dispersed lipids affected the crystalline arrangement, thus reducing the degree of crystallinity and promoting the formation of smaller crystals, as also deduced from RDX analyses. In contrast, polymer B behaved in the opposite way; Tm decreased from 183 °C in the first heating to 168 °C, which could be due to polymer changes occurring above 230 °C (degradation event). However, the presence of lipids in the matrix increased the Tm to 175 °C. The different effect of lipids on the material may lie in the different degree of lipid-polymer affinity. In polymer B, the interactions with lecithin lipids seemed to enhance the crystal size, while carvacrol promoted crystallinity, as observed in DRX analyses.

### 3.3. Functional Properties

The functional properties of the films are closely related to the molecular arrangement. Consequently, the chemical nature of the polymer and other included compounds, as well as their molecular interactions, play a crucial role in developing materials with specific requirements [48]. The molecular weight and the degree of hydrolysis of PVA had a marked influence on the tensile parameters (elastic modulus: EM, tensile strength: TS, and elongation: E, at break) and on the barrier properties (water vapour permeability: WVP; oxygen permeability: OP) of the films, as shown in Table 3. 

The films that were obtained with polymer A exhibited better mechanical properties (higher values of elastic modulus, resistance to break, and stretchability) than those that were obtained from polymer B. This has been attributed to the greater ability of the longer, deacetylated chains to form inter-chain hydrogen bonds. In contrast, the acetyl groups in polymer B interrupted the hydrogen bond formation in the shorter chains, which resulted in less cohesive matrices with lower crystallinity and, subsequently, reduced tensile performance. Restrepo et al. [49], also found improved mechanical properties of the PVA materials when the molecular weight and degree of hydrolysis increased. The barrier properties of the films also reflected these effects. Thus, the films of polymer A, with greater structural cohesion and crystallinity, exhibited better barrier capacity to both water vapour and oxygen than those of polymer B. 

The inclusion of liposomes, loaded or not with carvacrol, reduced the elastic modulus and resistance to break (TS) of films A. The presence of a dispersed phase in the matrix, due to the low lipid-polymer affinity, interferes in the polymer chain association, thus weakening the matrix cohesion forces [35]. In sample AL-CA5, with a more heterogeneous microstructural arrangement and lower crystalline fraction (25%), this weakening effect was more marked. In contrast, films B exhibited different behaviour, depending on the carvacrol load of liposomes. In all cases, liposomes reduced the elastic modulus of the films due to the weakening effect of the dispersed phase in the matrix cohesion forces, thus resulting in less rigid materials. Sapper et al. also reported this effect [15] for starch-gellan films with liposomes loaded or not with thyme essential oil. However, resistance and elongation at break increased in line with the carvacrol ratio, with the films becoming more stretchable. This agrees with the specific interactions of the acetyl group with CA that facilitate the slippage of the chains during the film stretching, delaying the break. Tongnuanchan, Benjakul, and Prodpran [50] reported that some compounds in essential oils might be able to cross-link chains, thereby improving the tensile properties. In this sense, some authors reported the increase in TS when cinnamon essential oil was incorporated into the soy protein isolate films [51] or chitosan films [52]. 

Barrier properties were also modified by including liposomes in the matrices. In both polymers, WVP slightly increased when liposomes were incorporated, whereas more notable changes were observed for OP. Different factors affected the molecular permeability in the matrix: the molecular diffusion of permeant that is affected by the microstructural features, such as the degree of crystallinity and the tortuosity factor that was introduced by the dispersed phase, and the molecular solubility in the matrix, affected by the chemical affinity of the permeant with the matrix components. In this sense, water molecules are highly soluble in the polymer continuous phase and less soluble in the lipid dispersed phase, whereas oxygen molecules behave in the opposite way. Thus, the balance of the different factors can explain the changes that were induced by liposomes in the barrier capacity of the films. The structural differences introduced by the dispersed phase only caused a small increase in the WVP values, given the high water solubility in the PVA continuous phase. In contrast, lipid incorporation enhanced the oxygen solubility in the matrix, promoting the increase in the OP values, additionally to that provoked by the structural changes. Samples AL-CA5 and BL, in particular, exhibited the lowest barrier capacity against both water vapour and oxygen in line with their lower crystalline fraction and greater microstructural heterogeneity, when compared with the other samples of the respective polymer. The incorporation of carvacrol-loaded liposomes gave rise to lower OP values in films B than non-loaded liposomes, which could be attributed to the greater crystallinity induced by carvacrol in these films (Figure 2). In polymer A, this effect of carvacrol was only observed for sample AL-CA10, which exhibited a greater dispersed phase with lower particle size, all of which implies a higher tortuosity factor for mass transfer. 

Table 4 shows the colour parameters (lightness (L*), Chroma (C_ab_*), and hue (h_ab_*)), and internal transmittance values at 460 nm (Ti), used as a transparency indicator, of the different samples. In general, the lightness values were lower in films A than in films B, while hue was higher in A, which was probably due to the different refractive indices of the matrices, associated with the corresponding microstructural arrangement affecting the light interactions. The incorporation of liposomes, loaded or not with CA, significantly reduced the lightness and hue and increased chrome (more saturated colour) of films A and B, in line with the colour imparted by lecithin [15,35]. In addition, the Ti values of the films with liposomes decreased, especially in films A. Similar effects were observed by other authors [35,53], when lecithin liposomes were incorporated into corn starch-sodium caseinate films.

## 4. Conclusions

The incorporation of liposome-encapsulated carvacrol into PVA films notably affected the microstructure of the films, their thermal behaviour, and their functional properties as packaging material, depending on the molecular weight and degree of hydrolysis of the polymer. Liposome-encapsulated carvacrol was better integrated in the polymer matrices of low M_W_, partially acetylated PVA, generating a more homogenous structure, where carvacrol enhanced the degree of crystallinity, whereas liposomes reduced the higher crystallinity in high-M_w_, non-acetylated PVA. A thermal protective effect of acetyl groups was observed in PVA, which was also observed when lecithin was incorporated as liposomes into the films. This effect shifted the thermodegradation temperature of PVA towards higher values, which were above its melting temperature, thus making it possible for polymer thermoprocessing to obtain films by means of the usual thermoplastic industrial process. Likewise, the presence of acetyl groups in the chain promoted chemical affinity with less polar compounds, such as carvacrol or lecithin components, permitting a greater carvacrol retention in the matrix and, thus, a more effective way of obtaining active films for food packaging. Liposome encapsulation also promoted the retention of carvacrol in the films, as compared with that incorporated as a non-encapsulated compound. Carvacrol-loaded liposomes reduced the elastic modulus and tensile strength of the PVA films, more markedly in the non-acetylated polymer, in line with the different changes in crystallinity and microstructure. Likewise, liposomes increased the oxygen permeability of the films according to the introduced structural modifications, but they maintained adequate values for food packaging applications. In general, the acetylated PVA exhibited better capacity as a carrier of carvacrol, encapsulated in liposomes, with great potential for the production of active films for food packaging applications.

## Figures and Tables

**Figure 1 polymers-12-00497-f001:**
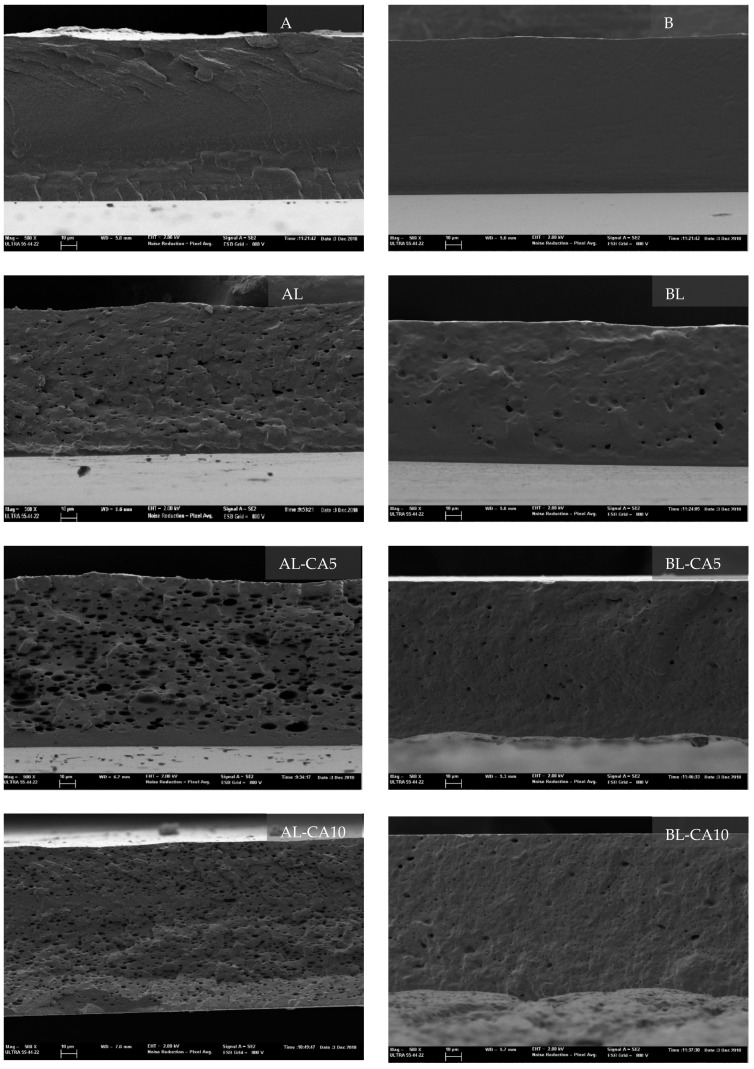
Field Emission Scanning Electron Microscope (FESEM) micrographs of the cross-section of the poly (vinyl) alcohol (PVA) A and B films with lecithin liposomes (L) and carvacrol loaded liposomes (L-CA) (5 or 10 g/100 g PVA) (magnification: 500X; bar: 10 μm).

**Figure 2 polymers-12-00497-f002:**
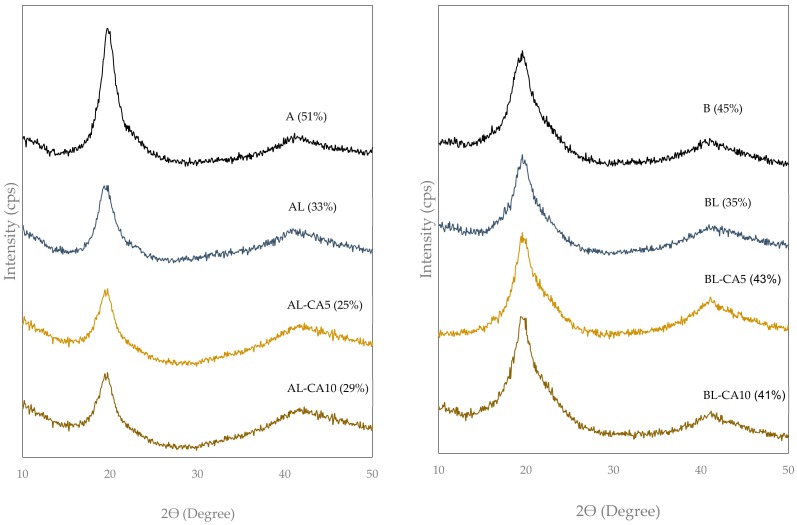
X-Ray diffraction spectra of the PVA films (A: left and B: right) without and with carvacrol (5 or 10 g/100 g PVA) previously encapsulated in lecithin liposomes. Percentages of crystallinity are shown for each sample.

**Figure 3 polymers-12-00497-f003:**
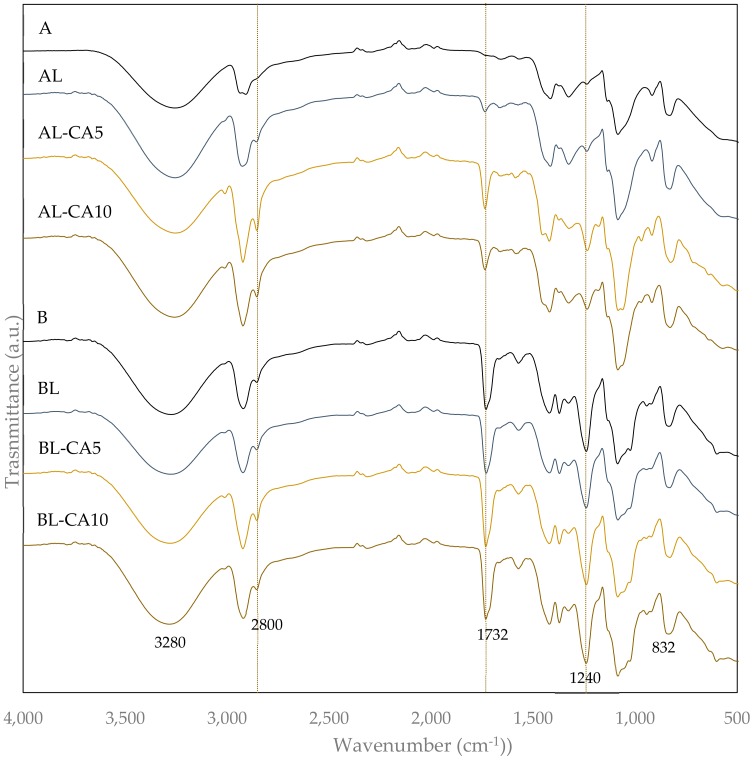
FTIR spectra of the PVA films (A and B) without and with carvacrol (5 or 10 g/100 g PVA) previously encapsulated in lecithin liposomes (L).

**Figure 4 polymers-12-00497-f004:**
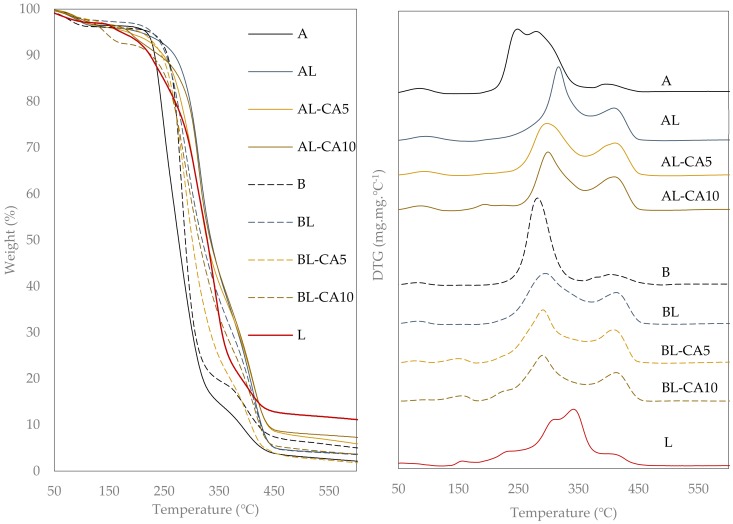
Thermogravimetric analysis (TGA) (left) and DTGA (right) curves of the PVA films (A and B) without and with carvacrol (5 or 10 g/100 g PVA) previously encapsulated in lecithin liposomes (L).

**Figure 5 polymers-12-00497-f005:**
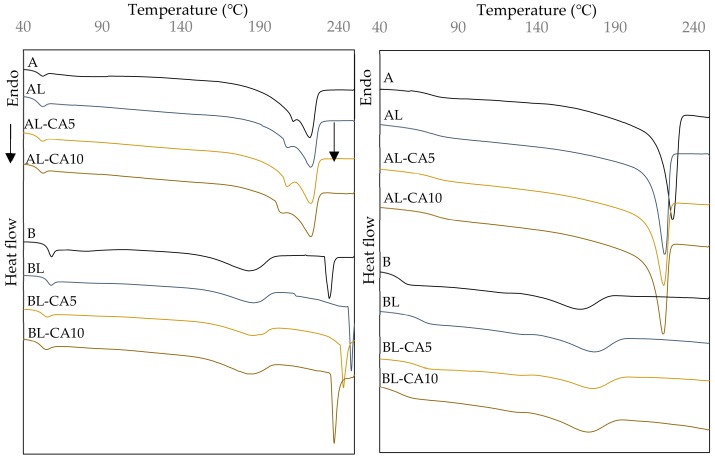
Differential scanning calorimetry (DSC) curves of the PVA films (A and B) with and without carvacrol (5 or 10 g/100 g PVA) previously encapsulated in lecithin (L). On the left, the first heating scan and on the right the second heating scan.

**Table 1 polymers-12-00497-t001:** Nominal mass fraction (x) of the different components of the films, extracted carvacrol in the final film, and its retention percentage. Mean values and standard deviation.

Sample	X_PVA_	X_CA_	X_L_	Extracted CA		CA-Retention (%)
				(mg CA/g PVA)	(mg CA/g Dry Film)	
A	1	-	-	-	-	-
AL	0,91	-	0,09	-	-	-
AL-CA5	0,87	0,04	0,09	28± 1	24± 1	55 ± 3^a^
AL-CA10	0,84	0,08	0,08	57± 2	48± 2	57 ± 30^a^
B	1	-	-	-	-	-
BL	0,91	-	0,09	-	-	-
BL-CA5	0,87	0,04	0,09	37± 2	32± 2	74 ± 2^c^
BL-CA10	0,84	0,08	0,08	67± 2	61± 2	67 ± 3^b^

Different superscript letters within the same column indicate significant differences among films (*p* < 0.05).

**Table 2 polymers-12-00497-t002:** Glass transition and melting temperature and enthalpy of the PVA films (A and B).

Sample	First Heating Scan			Second Heating Scan		
	Tg	Tm1	∆Hm (J/g PVA)	Tg	Tm	∆Hm (J/gPVA)
A	46,1 ± 0,2^a^	225 ± 5^a^	79 ± 1^b^	72 ± 2^cd^	225 ± 1^d^	73 ± 2^c^
AL	47,4 ± 0,5^b^	222 ± 1^a^	97 ± 6^c^	82 ± 1^e^	221 ± 1^c^	71 ± 8^bc^
AL-CA5	48,9 ± 0,1^c^	223 ± 2^a^	116 ± 7^d^	78 ± 2^de^	221 ± 1^c^	77 ± 2^c^
AL-CA10	49,0 ± 0,6^cd^	223 ± 2^a^	115 ± 13^d^	79 ± 2^de^	220 ± 1^c^	65± 10^b^
B	53,8 ± 0,4^e^	183 ± 1^a^	55 ± 2^a^	56 ± 3^a^	168 ± 1^a^	25 ± 2^a^
BL	54,3 ± 0,7^e^	186 ± 1^a^	54 ± 13^a^	64 ± 4^bc^	176 ± 1^b^	34 ± 4^a^
BL-CA5	50,0 ± 0,8^d^	185 ± 1^a^	53 ± 7^a^	58 ± 9^ab^	174 ± 2^b^	25 ± 1^a^
BL-CA10	48,9 ± 0,4^c^	186 ± 1^a^	54 ± 5^a^	58 ± 4^ab^	174 ± 2^b^	28 ± 2^a^

Different superscript letters within the same column indicate significant differences among films (*p* < 0.05).

**Table 3 polymers-12-00497-t003:** Tensile parameters (Tensile strength (TS), percentage elongation (E) and elastic modulus (EM)) and barrier properties (water vapour permeability, WVP; oxygen permeability, OP of the films. Mean values and standard deviation, in brackets.

Sample	Thickeness (µm)	TS (MPa)	E (%)	EM (MPa)	WVP x 10^3^ (g/m. h. kPa)	OP x 10^8^ (cm^3^/m. h. kPa)
A	101 ± 2^b^	153 ± 8^f^	135 ± 6^d^	80 ± 4^d^	2,47 ± 0,06^a^	0,38 ± 0,01^a^
AL	134 ± 2^d^	131 ± 7^e^	138 ± 5^d^	65 ± 8^C^	2,90 ± 0,30^b^	2,52 ± 0,24^b^
AL-CA5	131 ± 2^d^	111 ± 10^d^	137 ± 6^d^	67 ± 4^c^	3,60 ± 0,40^c^	5,47 ± 0,04^c^
AL-CA10	132 ± 2^d^	132 ± 12^e^	142 ± 8^d^	63 ± 5^c^	3,30 ± 0,02^bc^	1,72 ± 0,03^b^
B	95 ± 2^a^	44 ± 6^ab^	97 ± 6^b^	54 ± 5^b^	2,90 ± 0,02^b^	0,53 ± 0,05^a^
BL	122 ± 2^c^	40 ± 4^a^	86 ± 5^a^	43 ± 2^a^	3,50 ± 0,20^c^	16,10 ± 0,90^f^
BL-CA5	121 ± 2^c^	53 ± 5^b^	119 ± 4^c^	42 ± 2^a^	3,00 ± 0,30^b^	7,45 ± 0,08^d^
BL-CA10	124 ± 2^c^	71 ± 3^c^	140 ± 2^d^	40 ± 2^a^	3,10 ± 0,10^b^	10,3 ± 0,75^e^

Different superscript letters within the same column indicate significant differences among films (*p* < 0.05).

**Table 4 polymers-12-00497-t004:** Lightness (L *), chrome (Cab *), hue (hab *), and internal transmittance values at 460 nm (Ti) of the of the PVA films. Mean values and standard deviation.

Sample	L*	Cab*	hab*	T_i_
				(460 nm)
A	88 ± 2^C^	3 ± 1^a^	114 ± 11^e^	0,86 ± 0,01^e^
AL	78 ± 1^a^	10,8 ± 0,6^cd^	99 ± 2^ab^	0,82 ± 0,01^ab^
AL-CA5	78 ± 2^a^	8,7 ± 0,9^b^	105 ± 2^d^	0,82 ± 0,01^bc^
AL-CA10	78 ± 2^a^	10,6 ± 0,9^c^	100 ± 1^bc^	0,81 ± 0,01^a^
B	92 ± 1^d^	3,4 ± 0,5^a^	104 ± 2^cd^	0,86 ± 0,01^e^
BL	81 ± 1^b^	11,1 ± 0,6^cd^	96 ±1^a^	0,84 ± 0,01^d^
BL-CA5	81,8 ± 0,3^b^	11 ± 1^d^	98 ± 1^ab^	0,84 ± 0,01^d^
BL-CA10	81,0 ± 0,9^b^	13 ± 1^e^	96 ± 1^ab^	0,83 ± 0,01^c^

Different superscript letters within the same column indicate significant differences among films (*p* < 0.05).

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
