# Peer review of "The Incorporation of Carvacrol into Poly (vinyl alcohol) Films Encapsulated in Lecithin Liposomes"

_polymers, 2020, doi:10.3390/polym12020497_

Round 1

Reviewer 1 Report

Fig. 1, legend has to be corrected. As I assume the last pictures are for AL-CA10 and BL-CA10. Also, as always in SEM images, scale needs to be presented. It is not visible now and it seems that it differs between images.
Why there are differences in relationships of crystallinity determined by XRD and delta Hm, which also is used to determine the crystallinity between samples A and B? It should be adressed.
When describing mechanical properties Authors are writing that samples with polymer B show lower crystallinity, but XRD results state the opposite? Which statement is true then?
When describing the mechanical performance, Authors should refer to the porosity of materials.

Author Response

The authors appreciate their valuable contribution to improving the quality of the article paper.

  1. Fig. 1, legend has to be corrected. As I assume the last pictures are for AL-CA10 and BL-CA10. Also, as always in SEM images, scale needs to be presented. It is not visible now and it seems that it differs between images. 

Response: Figure 1 has been corrected. Scale is the same in all micrographs (bar: 10 mm). This has been included in the figure caption.

  1. Why there are differences in relationships of crystallinity determined by XRD and delta Hm, which also is used to determine the crystallinity between samples A and B? It should be addressed.

Response: Determination of crystallinity from DHm values has problems associated with the crystallization of the material during the thermal scanning, which could provide crystallinity values higher than that present in the material, which can be determined by XDR. Additionally, in the studied samples, other endothermic events, besides melting, occurred in the first heating scan (commented on the text), which can affect also the crystallinity values when determined from DSC analyses.

  1. When describing mechanical properties Authors are writing that samples with polymer B show lower crystallinity, but XRD results state the opposite? Which statement is true then?

Response: When PVA films did not contain liposomes, crystallinity of A polymer was higher than that of B polymer. As discussed in the text, the incorporation of non-loaded liposomes reduced crystallinity of both polymer matrices. However, when liposomes were loaded with carvacrol, different effects were observed for A and B matrices. Crystallinity of B matrix containing liposomes was promoted by the presence of carvacrol, whereas this did not occur in A samples. Thus, depending on the presence or not of carvacrol, the crystallinity of A and B matrices can vary.

  1. When describing the mechanical performance, Authors should refer to the porosity of materials.

Response: The apparent pores in the polymer matrices when liposomes were incorporated correspond to the dispersed lipid phase (lecithin associations with or without carvacrol, which are not completely miscible with the polymer chains), which is referred in the text when mechanical performance of the material was discussed (lines 441-447).

Reviewer 2 Report

The paper is very interesting and the results are presented in a clear way, as well as materials and methods are well described.

However, I think it would be important to include in this paper, the description of the release kinetic of CA in both types of PVA studied, in order to select adequately the film in food preservation.

The references are adecuate and show the great experience of the group in this type of studies.

Author Response

The authors appreciate their valuable contribution to improving the quality of the article paper.

  1. The paper is very interesting and the results are presented in a clear way, as well as materials and methods are well described.

However, I think it would be important to include in this paper, the description of the release kinetic of CA in both types of PVA studied, in order to select adequately the film in food preservation.

The references are adequate and show the great experience of the group in this type of studies.

Response: Effectively, the release kinetics of carvacrol from the different polymer matrices in food simulants of different polarity is an important feature, to know the effectiveness of the films as antimicrobial material for food packaging and this will be carried out in further studies, together with the antimicrobial studies in different food substrates. 

Reviewer 3 Report

This work is about the fabrication of PVA active films incorporated with lecithin-encapsulated carvacrol. The influence of Mw and DH of PVA on its ability to retain carvacrol has been analysed, as well as the changes in the film microstructure, thermal behaviour and functional properties provoked by liposome incorporation. Unfortunately, the antimicrobial and antioxidant properties of films were not evaluated. In addition, the lecithin incorporated with carvacrol was not characterized. Some suggestions were given as follows:

1.The lecithin incorporated with carvacrol should be characterized.

2. The antimicrobial and antioxidant properties of films and the release behavior of carvacrol should be evaluated

3. Figure 1, check the last two photographs: AL-, BL-?; Figure 2, which is for A? B?; Figure 3, the characteristic bands should be marked; Figure 4, the curves in TGA (left) were not indicated; Figure 5, the curves in TGA (left) were not indicated, additionally, the quality is poor.

4. What are the interactions between lecithin-encapsulated carvacrol and PVA matrix?

5. Figure 4, the weight loss rate (%) should be illustrated. What is the residual?

6.As indicated, the TS and E were both enhanced with the increase of CA, why?

7.Check the style of references carefully

Author Response

Response to Reviewer 3 Comments

The authors appreciate their valuable contribution to improving the quality of the article paper.

This work is about the fabrication of PVA active films incorporated with lecithin-encapsulated carvacrol. The influence of Mw and DH of PVA on its ability to retain carvacrol has been analysed, as well as the changes in the film microstructure, thermal behaviour and functional properties provoked by liposome incorporation. Unfortunately, the antimicrobial and antioxidant properties of films were not evaluated. In addition, the lecithin incorporated with carvacrol was not characterized. Some suggestions were given as follows:

  1. The lecithin incorporated with carvacrol should be characterized.

Response: Composition of lecithin supplied by the manufacturer has been included in the text (Lines 79-81)

  1. The antimicrobial and antioxidant properties of films and the release behavior of carvacrol should be evaluated

Response: The information about the antimicrobial and antioxidant properties and release kinetics of carvacrol is relevant and very important, and will be analysed in further studies. This paper aims to analyse the influence of the molecular weight (Mw) and degree of hydrolysis of PVA on its ability to incorporate carvacrol, liposome encapsulated, and its influence on the  film microstructure, crystallinity, thermal behaviour and functional properties as packaging material (mechanical and barrier properties).

  1. Figure 1, check the last two photographs: AL-, BL-?; Figure 2, which is for A? B?; Figure 3, the characteristic bands should be marked; Figure 4, the curves in TGA (left) were not indicated; Figure 5, the curves in TGA (left) were not indicated, additionally, the quality is poor.

Response:

Figure 1 was corrected specifying the sample code.

Figure 2 was corrected, indicating samples A and B.

Figure 3 was modified incorporating the marks of characteristic bands.

Figure 4. TGA curves has been overlapped without shift in the Y-axe. All curves have the same origin with 100 % of initial mass and the residual mass can be compared for the different samples.

Figure 5 was corrected by adding marks to the left.

  1. What are the interactions between lecithin-encapsulated carvacrol and PVA matrix?

Response: As discussed in the text, different kinds of interactions between polymer chains, lecithin and carvacrol could be established in the matrices depending on the presence or not of acetyl groups in the PVA chains. While polymer A is highly hydrophilic with a high number of hydroxyl groups, the presence of residual acetyl groups in the polymer B chains provides them with an amphipathic nature that promotes the chemical affinity between the polymer and the lipid compounds. Particularly, specific interactions between the carvacrol released from liposomes with acetylated PVA were also observed, and described as the formation of Lewis adducts between the carbonyl and the phenolic hydroxyl. Likewise, specific interactions between carvacrol and lecithin were also commented in the text, resulting in a reduction of the repulsive forces among the head groups of phospholipids in the liposomes.   

  1. Figure 4, the weight loss rate (%) should be illustrated. What is the residual?

Response: Figure 4 has been changed according to the reviewer suggestion.

  1. As indicated, the TS and E were both enhanced with the increase of CA, why?

Response: These effects have been observed only for B polymer, in agreement with the specific interactions of the acetyl group with carvacrol that facilitate the slippage of the chains during the film stretching, delaying the break, and giving rise to a more stretchable material, whose resistance to break increase in line with the deformation level. These specific interactions have been already described in the “Microstructure section” and the carvacrol retention capacity of polymer B. As previously commented on, interactions are based on the formation of Lewis adducts between carvacrol and the residual acetyl groups of the PVA B chains.

  1. Check the style of references carefully

Response: This aspect has been reviewed.

Round 2

Reviewer 3 Report

The manuscript has been improved to some extent.